# Culturable Endophyte Fungi of the Well-Conserved Coastal Dune Vegetation Located on the East Coast of the Korean Peninsula

Jong Myong Park [1,2] and Young-Hyun You [3,*]

1    Incheon Metropolitan City Institute of Public Health and Environment, Incheon 22320, Republic of Korea; eveningwater@korea.kr
2    Water Quality Research Institute, Waterworks Headquarters, Incheon 21316, Republic of Korea
3    Biological Resources Utilization Division, National Institute of Biological Resources, Incheon 22689, Republic of Korea
*    Correspondence: rocer2404@korea.kr; Tel.: +82-55-350-7232

**Abstract:** This study aimed to determine the diversity and distribution of endophytic fungi in coastal dune vegetation. Endophytic fungi promote plant growth and protect host plants from environmental stress and pathogens. Plants that have flourished as a result are critical for protecting coastal sand dunes from erosion. A total of 361 root-colonizing fungal endophytes were purely isolated from 24 halophyte species comprising all dune plant communities indigenous to a well-conserved coastal area based on morphological differences. Molecular identification and phylogeny using amplified ribosomal RNA sequences and internal transcribed spacer regions identified that the fungal isolates belong to seven classes and 39 genera. *Penicillium* (43.21%) was the most dominant genera, followed by *Talaromyces* (16.90%) and *Aspergillus* (11.91%). Furthermore, these genera present a wide host range. However, 16 other genera exhibited strong host specificity. When compared to other herbaceous or shrub host plant species, *Talaromyces* predominated as endophytes of the roots of the canopy-forming coastal windbreak tree *Pinus thunbergii*. Based on Margalef's, Menhinick's, Shannon's, and Simpson's diversity indices, the root-colonizing endophytes of *P. thunbergii* had higher morphological diversity. The endophyte fungi associated with five of the coastal plants studied are heretofore unreported. In fact, of all fungal genera characterized here, 13 genera (30%) have not been previously reported as marine fungal endophytes or coastal fungi. The foregoing results suggest that future coastal sand dune conservation studies should examine the biological resources of entire bioclusters and not merely the dominant plants or their endosymbionts.

**Keywords:** coastal plant; coastal sand dunes; ecotone; fungus; halophyte; root-colonizing endophyte





## 1. Introduction

Coastal sand dunes tend to form on the landward side of natural beaches [1]. Coastal halophyte vegetation that develops in sand dunes serves as a natural embankment, protecting the coastline from erosion and sand loss [2–4]. Furthermore, halophytes can tolerate salt accumulation and may regulate the Na$^+$ concentration in the rhizosphere. As coastal vegetation effectively maintains sand dunes as buffer zones for the coast and the adjacent inland, these areas become ecotones and biodiversity repositories [5]. Dune habitats provide niches for highly specialized plant and animal species, including those that are rare or even endangered.

However, the widespread expansion of human populations is artificially destroying dunes. Anthropogenic activity includes land development for recreational purposes and alterations that prevent the encroachment of sand into inhabited areas [6]. Well-preserved dunes store underground water, protect the land from seawater and salt intrusion, and support the plant community. As a result, the destruction of sand dunes and coastal

vegetation threatens human habitation as well [7,8]. Massive natural or artificial sand loss and the destruction of native vegetation alter coastlines, narrow beaches, and cause erosion to intrude into coastal cities [9,10]. Additionally, the loss of halophytes removes salt elimination, which can lead to drought that may jeopardize agriculture and mesophytes remote from the coastline [11]. These negative effects inevitably result in serious losses of biological resources, tourism, natural scenery, cultural heritage, and coastal ecosystem diversity [7]. The Korean Peninsula has well-formed coastal sand dunes [9]. However, many of them have disappeared as a consequence of poorly managed development.

Endophytic microorganisms play key roles in the growth, reproduction, and abiotic stress tolerance of their halophyte hosts [12,13]. Diverse endophytes collections were reported to promote their host plant's growth and reproduction, as well as induce systemic resistance to plant pathogens or harsh environments, such as extreme marine conditions [14–16]. As a result, securing the culture collection of endophytic fungi native to halophyte species and identifying their ecological diversity at well-preserved sites is essential for restoring devastated coastal environments. Such a culture collection representative of the biodiversity of these areas would be of great value in light of the Nagoya Protocol and to secure the availability of biological resources [17]. Marine stressors include high salt concentrations, sudden and drastic temperature changes, sea winds and fog, and ultraviolet radiation reflected from the sea surface [18]. Coastal halophyte–endophytic fungus interactions can withstand these stressors and help host plants thrive under harsh marine conditions [19].

Well-established and thriving coastal plants conserve coastal sand dunes. Hence, important research objectives include compiling a culture collection of endophytes indigenous to well-conserved coastal sand dunes and characterizing their ecological diversity and host relationships. This information could then be applied towards restoring coastal dunes. Including these reasons, studies on endophytic fungi present in coastal dunes have been conducted for various purposes. Endophytic fungi profiles have been continuously reported for halophytes or coastal native plant species, with no previous report of fungal clusters [20–26].

The Korean Peninsula is surrounded by coastline on three sides [18,27]. Different dune types and plant communities are formed because of regional variations in climate zones and geographical factors. Here, we selected and studied well-preserved, restricted coastal sand dunes along the East Coast of the Korean Peninsula that are subjected to the most severe marine environmental conditions. We investigated them prior to their permanent opening to the general public. An aim of the present study was to identify the root-colonizing endophytic fungi in the coastal plant species comprising the dune vegetation rather than focusing on the dominant or indicator plant species. We characterized 24 coastal plant species constituting all regional sand dune vegetation based on their taxonomy. We isolated and identified their fungal endophytes according to their universal sequences. This work disclosed several heretofore unreported associations between endophytes and coastal plants. We also determined the distribution and diversity of endophytic fungi and compared them among host plant species.

## 2. Materials and Methods

### 2.1. Coastal Dune Selection and Site Descriptions

Each coast on the Korean Peninsula has unique topographical characteristics. The beaches of the east coast have thick sand and low sediment concentrations and are affected by severe salt intrusion and strong sea winds [28] that form and move dunes.

The coastline is monotonous because of the depth of the water [29]. For the present study, we selected the Hwajin sand dune as it is representative of the topographical characteristics of the east coast of the Korean Peninsula (Figure 1).

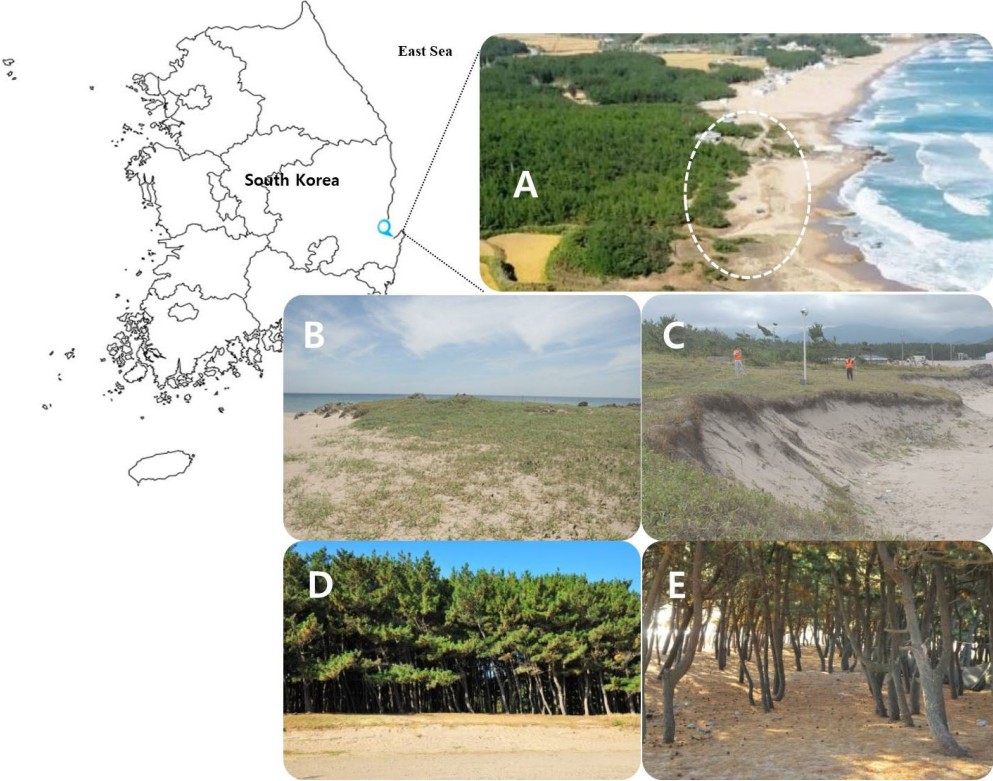

**Figure 1. Sampling site and plant ecology description**. (**A**) Hwajin Beach ecotone area. Expanded aerial photographs of the region could not be acquired or presented as the region is a military base. (**B**) Side view of dune from the north. (**C**) Side view of the dune from the south. (**D**) Border between herbaceous plants and *Pinus thunbergii* colony. (**E**) Interior of dune canopy showing *P. thunbergii* dominance.

The foot of the mountain at the rear of the sand dune has been closed to the public for the past 60 years because of the ceasefire on the Korean Peninsula. The sand dunes are included in the restricted areas. Hence, they are protected from development and anthropogenic disturbances. There has been no military activity in this zone for the past 50 years. The beach in front of the dunes is 400 m long, 100 m wide, and 1.5 m deep. Its coordinates are N 36°14′90″ and E 129°22′50″, and its administrative address is Hwajin-ri, Songmyeon, Buk-gu, Pohang, Gyeongsangbuk-do, Republic of Korea.

### 2.2. Plant Taxonomy and Sampling

The sampling targets were 24 coastal plant species native to the dunes and identified by a plant taxonomist using morphological criteria [30]. Roots were collected from flourishing, undisturbed communities of indigenous halophytic species. Fifteen individual plants of each species were harvested along with their proximal root-layer beach soil to minimize physiological changes. Sterile distilled water (SDW) and sterile 0.1% ($v/v$) Tween 80 solution (Sigma-Aldrich Corp., St. Louis, MO, USA) were sprayed onto the root surfaces to remove suspended solids and normal microflora [9]. The plants were submerged twice in 1.0% ($v/v$) perchloric acid ($HClO_4$) for 10 min each time and were then washed three or four times with SDW [9,31]. Residual water was removed with sterile, dry gauze. Fifty root pieces per plant sample were each cut to 3–4 cm length.

### 2.3. Genomic DNA Extraction and Polymerase Chain Reaction (PCR)

Pretreated samples were loaded into Hagem minimal medium containing 80 ppm streptomycin (Sigma-Aldrich Corp.) to exclude root bacteria. The samples were incubated at 25 °C for 15 d [22,32]. The endophytic fungi were subcultured for pure isolation using

the foregoing medium and conditions. The pure isolates were incubated on potato dextrose agar (Difco; Detroit, MI, USA) and selected according to their morphological differences.

The present study adopted culture-dependent methods in accordance with recent studies on root-colonizing endophytes in coastal terrain [9,25,33,34] and other environments [26,35–38]. The 361 endophytic fungi isolated from the 24 halophyte species were inoculated into potato dextrose broth (Difco) and incubated at 25 °C for 7 d on a rotary shaker at 120 rpm. Filtered mycobionts were lyophilized for 2 d. A DNeasy Plant Mini Kit (Qiagen, Hilden, Germany) was used to extract the genomic DNA from the lyophilized mycobionts.

Primers targeting the partial sequence of 18S ribosomal RNA (rRNA), the complete sequence of internal transcribed spacer (ITS) 1, 5.8S rRNA, the ITS2 region, and the partial sequence of 28S rRNA were used for amplification (ITS1, 5′–TCC GTA GGT GAA CCT GCG G–3′, and ITS4, 5′–TCC GCT TAT TGATAT GC–3′) [39]. The PCR amplification conditions were as follows: pre-denaturation at 94 °C for 4 min; denaturation at 94 °C for 1 min; annealing at 55–58 °C for 1 min; and extension at 72 °C for 2 min. There were 35 cycles followed by a final extension at 72 °C for 2 min [39]. The PCR products were confirmed by gel electrophoresis (1.5% ($w/v$) agarose gel stained with ethidium bromide). The band pattern was observed under a UV transilluminator. AccuPrep PCR and Gel Extraction Kit (Bioneer, Daejeon, Korea) were used to purify the PCR products. An ABI 3730XL DNA analyzer (Applied Biosystems, Waltham, MA, USA) was used for ITS sequencing.

### 2.4. Identification and Phylogeny

The 361 root-colonizing fungal isolates were partially identified based on signature sequence homology with data collected from the GenBank database [40]. The partial *ITS1* and *28S* sequences and the complete *5.8S* and *IST2* sequences obtained from the fungal isolates were compared against the data deposited in the GenBank database. The BLASTn tool (https://blast.ncbi.nlm.nih.gov/Blast.cgi?PROGRAM=blastn&PAGE_TYPE=BlastSearch&BLAST_SPEC=&LINK_LOC=blasttab&LAST_PAGE=blastn) was used to determine their taxonomical relationships. Evolutionary analyses were conducted using Molecular Evolutionary Genetics Analysis (MEGA) v. X (64 bit for Windows) [41,42] with sequence alignments prepared using Clustal W v. 2.0.10 [43,44]. Evolutionary history inferred from evolutionary relationships between taxa was assessed using the minimum evolution method [45]. 'Associated' indicated taxa clustered together in a percentage of the replicate trees in the bootstrap test (1000 replicates) [46]. Evolutionary distances were computed by the Tajima-Nei method [47] and expressed according to the number of base substitutions per site. The minimum evolution tree was explored with the close-neighbor interchange algorithm [48] at search level = 1. A neighbor-joining (NJ) algorithm [49] was used to generate the initial tree. All ambiguous positions for each sequence pair were removed using the pairwise deletion option. The 361 fungal signature sequences obtained here were deposited in GenBank under the assigned accession numbers (Supplementary Table S1).

### 2.5. Biodiversity Assessment

Fungal diversity was compared among halophyte species [50]. Genus-level diversity was determined using (1) Margalef's richness [51], (2) Menhinick's index [52], (3) Shannon's diversity (H′) [53], and (4) Simpson's indices (1-D) [53,54].

## 3. Results and Discussion

### 3.1. Plant Taxonomy and Distribution

Twenty-four coastal dune plant species comprising all regional vegetation were collected and identified. A windbreak tree (*Pinus thunbergii*) forest formed the back edge of the coastal dunes and well-conserved vegetation formed the canopy. The dunes and beach were 100 m wide from the edge of the *P. thunbergii* canopy to sea level (Figure 1). All coastal plant species were morphologically classified in the phylum Streptophyta, classes Magnoliopsida and Pinopsida, 12 orders, 15 families, and 23 genera (Table 1).

**Table 1.** Classification and identification of halophytes native to coastal dunes.

| Scientific Name | Taxonomic Information | | | Life Cycle | |
| --- | --- | --- | --- | --- | --- |
| | Class | Order | Family | | |
| *Cnidium japonicum* | | Apiales | Apiaceae | | biennial |
| *Artemisia scoparia* | | Asterales | Asteraceae | | perennial |
| *Spergularia marina* | | Caryophyllales | Caryophyllaceae | | biennial |
| *Cerastium fischerianum* | | | Caryophyllaceae | | annual |
| *Atriplex gmelinii* | | | Chenopodiaceae | herbaceous | perennial |
| *Lysimachia mauritiana* | | Ericales | Primulaceae | | biennial |
| *Lathyrus japonicus* | | Fabales | Fabaceae | | perennial |
| *Scutellaria strigillosa* | | | | | |
| *Vitex rotundifolia* | | Lamiales | Lamiaceae | xylophyte frutescence (low-bush type) | perennial |
| *Linaria japonica* | | | Plantaginaceae | | |
| *Plantago camtschatica* | Magnoliopsida | | Plantaginaceae | | |
| *Carexkobomugi* | | | Cyperaceae | | perennial |
| *Carex pumila* | | | Cyperaceae | | |
| *Elymus mollis* | | | | herbaceous | |
| *Ischaemum anthephoroides* | | Poales | Poaceae | | |
| *Phragmites australis* | | | Poaceae | | |
| *Setaria viridis* | | | | | annual |
| *Zoysia sinica* | | | | | perennial |
| *Cynodon dactylon* | | | | | |
| *Rosa rugosa* | | Rosales | Rosaceae | xylophyte frutescence (low-bush type) | perennial |
| *Sedum kamtschaticum* | | Saxifragales | Crassulaceae | | |
| *Aster sphathulifolius* | | Asterales | Asteraceae | herbaceous | perennial |
| *Sedum oryzifolium* | | Saxifragales | Crassulaceae | | |
| *Pinus thunbergii* | Pinopsida | Pinopsida | Pinaceae | xylophyte, tree | perennial |

The perennial trees *Vitex rotundifolia* (frutescence; low-bush type), *Rosa rugosa* (frutescence; low-bush type), and *P. thunbergii* (tall evergreen tree, needle-leaf) were identified based on the growth cycles of the coastal plants. Among the herbaceous plants, 16 perennials, 3 biennials, and 2 annuals (Table 1) were collected. All the foregoing species except *P. thunbergii* belonged to the class Magnoliopsida (angiosperms). There were eight species in order Poales, four in order Lamales, and three in order Caryophyllales (Table 1). There were six species in the family Poaceae.

Quantitative dominance was determined by the area, population sizes, and colonies. *Carex kobomugi* (herbaceous sedge; perennial), *V. rotundifolia* [perennial; round-leaf vitex; tree; (low-bush type)], *R. rugosa* [perennial; beach rose; tree; (low-bush type)], *P. thunbergii* (perennial; black pine; tree), and *Ischaemum anthephoroides* (herbaceous; murainagrass; perennial) were the dominant species at the rear of the dunes while *P. thunbergii* formed a canopy forest.

There was no stratification of the tree-shrub-herbaceous order. Certain halophyte colonies were clearly distinct from others. The *P. thunbergii* canopy separated the coast and

inland areas, and its lower part was completely shaded. As a result, herbaceous plants in the interior of the canopy were excluded throughout the year.

### 3.2. Endophytic Fungal Taxonomy and Phylogeny

Analysis of the signature sequences of 361 root-colonizing fungal cultures isolated in pure culture revealed various species in seven classes (Agaricomycetes, Ascomycetes, Dothideomycetes, Eurotiomycetes, Leotiomycetes, Saccharomycetes, and Sordariomycetes), 20 orders, 33 families, and 39 genera (Table 2). The signature sequences of each isolate shared the highest similarity with GenBank sequences, and there were 134 type strains (Table 2). The use of the ITS and rRNA regions alone does not definitively identify fungi at the species level. Partial molecular identification of the entire culture collection was evaluated by phylogenetic tree construction (Figure S1).

**Table 2.** Taxonomic identity of endophytes isolated from each host halophyte species.

| Host Species | Taxonomic Endophyte Information | | | | | |
|---|---|---|---|---|---|---|
| | Class | Order | Family | Genus | Species | Total Isolates |
| *C. japonicum* | 4 | 5 | 5 | 6 | 18 | 25 |
| *A. scoparia* | 4 | 6 | 7 | 8 | 12 | 18 |
| *S. marina* | 4 | 4 | 5 | 6 | 8 | 10 |
| *C. fischerianum* | 4 | 5 | 6 | 6 | 12 | 16 |
| *A. gmelinii* | 4 | 4 | 5 | 6 | 27 | 31 |
| *L. mauritiana* | 2 | 2 | 2 | 3 | 3 | 3 |
| *L. japonicus* | 2 | 2 | 2 | 2 | 4 | 5 |
| *S. strigillosa* | 2 | 3 | 4 | 5 | 11 | 17 |
| *V. rotundifolia* | 2 | 2 | 2 | 2 | 3 | 3 |
| *L. japonica* | 4 | 4 | 4 | 5 | 9 | 10 |
| *P. camtschatica* | 5 | 6 | 8 | 10 | 16 | 19 |
| *C. kobomugi* | 2 | 2 | 2 | 4 | 7 | 7 |
| *C. pumila* | 4 | 5 | 8 | 8 | 10 | 11 |
| *E. mollis* Trin. | 4 | 4 | 4 | 5 | 8 | 8 |
| *I. anthephoroides* | 3 | 3 | 5 | 8 | 20 | 31 |
| *P. australis* | 4 | 5 | 6 | 6 | 6 | 6 |
| *S. viridis* | 3 | 4 | 5 | 6 | 14 | 17 |
| *Z. sinica* | 3 | 4 | 5 | 6 | 7 | 8 |
| *C. dactylon* | 5 | 5 | 5 | 6 | 8 | 10 |
| *R. rugosa* | 4 | 4 | 5 | 5 | 7 | 18 |
| *S. kamtschaticum* | 4 | 3 | 4 | 6 | 10 | 15 |
| *A. sphathulifolius* | 5 | 6 | 9 | 8 | 16 | 23 |
| *S. oryzifolium* | 3 | 3 | 4 | 6 | 11 | 14 |
| *P. thunbergii* | 3 | 4 | 5 | 8 | 19 | 36 |

### 3.3. Fungal Distribution in Coastal Dune Vegetation

3.3.1. Dominance of Specific Taxa

The root-colonizing endophytic fungi in the dune vegetation were dominated by species in the class Sordariomycetes (52.09%). Other frequently encountered classes were Eurotiomycetes (29.36%) and Dothideomycetes (11.63%) (Figure 2). Order Hypocreales was

the most frequently encountered (49.03%), followed by Eurotiales (31.30%) and Pleosporales (8.86%). The Family Nectriaceae (46.26%) predominated, and Trichocomaceae (29.09%) and Phaeosphaeriaceae (4.16%) were also represented. *Penicillium* and *Talaromyces* accounted for 43.21% and 16.90% of all fungal species, respectively. *Aspergillus* (11.91%) was also frequently observed. Other genera contributed to ≤3–4% of all fungal species (Figure 2). *Aspergillus, Penicillium,* and *Talaromyces* distribute marine terrains or solubilize rock in them [37,55–59].

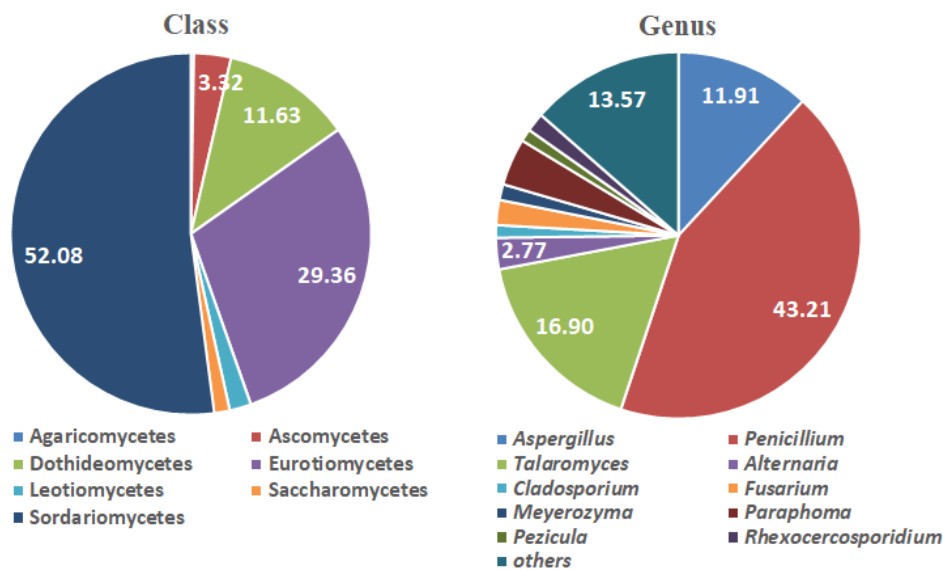

**Figure 2.** Fungal class and genus distributions in all halophyte clusters.

However, the Sordariomycetes do not predominate in all halophytes. The overall dominance pattern of root-colonizing endophytes derived from dune plant species does not correspond to that for each host plant species (Figure 3). Sordariomycetes were most prevalent in *Atriplex gmelinii* (87.1%), *Aster sphathulifolius* (69.6%), *Cerastium fischerianum* (75.0%), *Carex kobomugi* (71.4%), *Elymus mollis* (50.0%), *Lathyrus japonicus* (80%), *Plantago camtschatica* (52.6%), *Sedum kamtschaticum* (66.7%), *Sedum oryzifolium* (57.1%), *Scutellaria strigillosa* (58.8%), and *Zoysia sinica* (50.0%). By contrast, the host plant species for the other fungal classes were *Cynodon dactylon* and *Linaria japonica* (Eurotiomycetes; approximately 50%), *Lysimachia mauritiana* (Eurotiomycetes; > 60%), *Phragmites australis* (Dothideomycetes; > 50%), *P. thunbergii, R. rugosa, Spergularia marina,* and *V. rotundifolia* (Eurotiomycetes; > 60%). *Penicillium* was the most prevalent fungal genus in *A. gmelinii* (74.2%) and *A. sphathulifolius* (56.5%).

### 3.3.2. Host Ranges and Endophyte Distribution

*Penicillium* is commonly identified in a wide range of plant species. Nevertheless, it was not identified in *P. thunbergii* (Figure 3; Table S1). Certain host plants may be highly selective with *Penicillium* species. Selection pressure from the marine environment may influence the predominance of certain genera (Table 3).

Species-level diversity might have increased to a greater extent than genus-level diversity. It is difficult to classify fungi to the species level by partial identification with universal sequences [9,25,33,35–38]. However, the isolates were highly homogeneous, with various types of strains deposited in the GenBank databases (134 species; Table S1). The number of endophyte isolates resembling the type strain species listed in the NCBI increased for *Penicillium* and *Talaromyces*. As a result, fungal species richness or evenness may also be increased [60].

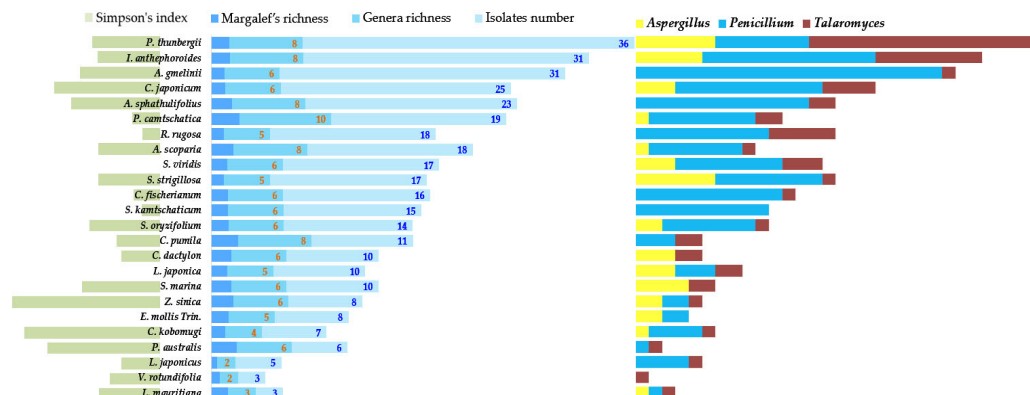

**Figure 3. Morphological diversity: richness (middle), evenness (left), and ratios of dominant genera (right).** Fungal richness and morphological diversity varied depending on the dominance of specific taxa. Predominance ratios were deduced to identify the genera with lower morphological diversity and abundance. Compared with other plant species, the richness of *P. thunbergii* was reduced by the unrivaled dominance of *Talaromyces*. Nevertheless, *P. thunbergii* had the widest morphological diversity, as more species were isolated from *Taralomyces* than any other fungal genera. Species diversity can be determined from the number of isolates within *Talaromyces*, as the strains were isolated on the basis of morphological differentiation. By contrast, *Aspergillus* predominated in *S. marina*, *L. japonica*, and *C. dactylon*, whereas *Penicillium* was dominant in the other host plant species.

**Table 3.** Fungal endophyte diversity.

| Host Plant | Genera Richness | Margalef's Richness | Menhinick's Index | Shannon's Diversity | Simpson's Index |
|---|---|---|---|---|---|
| *L. mauritiana* | 3 | 1.820 | 1.200 | 1.488 | 0.247 |
| *V. rotundifolia* | 2 | 0.910 | 1.886 | 1.687 | 0.203 |
| *L. japonicus* | 2 | 0.621 | 1.897 | 1.609 | 0.156 |
| *P. australis* | 6 | 2.791 | 1.500 | 1.124 | 0.458 |
| *C. kobomugi* | 4 | 1.542 | 1.078 | 0.973 | 0.551 |
| *E. mollis* Trin. | 5 | 1.924 | 1.732 | 1.099 | 0.000 |
| *Z. sinica* | 6 | 2.404 | 0.894 | 0.500 | 0.600 |
| *S. marina* | 6 | 2.171 | 1.213 | 1.222 | 0.316 |
| *L. japonica* | 5 | 1.737 | 1.155 | 0.732 | 0.000 |
| *C. dactylon* | 6 | 2.171 | 1.581 | 1.505 | 0.156 |
| *C. pumila* | 8 | 2.919 | 2.294 | 1.923 | 0.175 |
| *S. oryzifolium* | 6 | 1.895 | 1.512 | 1.154 | 0.286 |
| *S. kamtschaticum* | 6 | 1.846 | 2.412 | 1.972 | 0.073 |
| *C. fischerianum* | 6 | 1.803 | 1.768 | 1.560 | 0.107 |
| *S. strigillosa* | 5 | 1.412 | 1.437 | 1.562 | 0.249 |
| *S. viridis* | 6 | 1.765 | 2.449 | 1.792 | 0.000 |
| *A. scoparia* | 8 | 2.422 | 1.455 | 1.467 | 0.250 |
| *R. rugosa* | 5 | 1.384 | 2.121 | 1.473 | 0.071 |
| *P. camtschatica* | 10 | 3.057 | 1.897 | 1.696 | 0.111 |
| *A. sphathulifolius* | 8 | 2.233 | 1.179 | 1.164 | 0.359 |
| *C. japonicum* | 6 | 1.553 | 1.549 | 0.992 | 0.429 |
| *A. gmelinii* | 6 | 1.456 | 1.668 | 1.482 | 0.324 |
| *I. anthephoroides* | 8 | 2.038 | 1.604 | 1.468 | 0.253 |
| *P. thunbergii* | 8 | 1.953 | 1.333 | 1.530 | 0.275 |

The host plants also exhibited strong selectivity for certain fungal groups. The dominance of *Paraphoma* (4.16%) was lower than that of all other fungal genera in the dune vegetation (Table S1). However, several endophyte isolates strongly resembled the type strain *Paraphoma rhaphiolepidis* (Table S1). Hence, certain species within particular fungal genera may be strongly associated with a certain host plant species.

*Talaromyces* (22 hosts), *Penicillium* (21 hosts), and *Aspergillus* (15 hosts) were widely distributed among the dune plant species (Table S1). At least 23 of the 39 root-colonizing fungi were common to at least two host species (Table S1). Therefore, the dependence of these fungal genera on certain plant host species might be weak. Moreover, all the species in the foregoing genera may be adapted to harsh marine environments. However, the distribution of the type strains most closely resembling all isolates in the foregoing genera varied with host plant species (Table 3). Therefore, the interactions between individual plant and fungal species could be specific [61].

Though the three foregoing fungal genera were widely distributed among the dune plants, in certain cases, the species distribution was not extensive (Table 3). Sixteen of the 39 fungal genera were distinctively distributed among the host plants. While they were not the dominant taxa, they displayed high host species specificity or narrow host range.

Symbiotic relationships cannot be judged based on the host specificity (dependence) of particular fungal species. Common fungal plant pathogens show high dependence on specific hosts as well. For this reason, symbioses must be assessed using metadata from previous literature on fungal species and plant host interactions. The dominant fungal species *Aspergillus terreus* and *Penicillium thomii* interacted with seven and eight dune plant species, respectively (Table S1). Accurate identification by molecular methods is difficult above the genus level. However, type strains, including *P. thomii*, were identified in a wide range of host species. These isolates were confirmed to have a higher similarity. Endophytes from eight plant species (and particularly *Aster spathulifolius* and *S. oryzifolium*) had high similarity to *P. thomii*. *Penicillium thomii* has been detected in the deep sea [37] and coastal habitats [55], and reported as a symbiont in marine algae and plants [59]. It helps plants survive on poor sandy soil and solubilizes rock [56]. Future research should examine the impact of these species on plant settlement and reproduction in coastal dunes.

Here, isolates from the 11 coastal plant species (and especially *R. rugosa* and *I. anhephoroides*) were highly homogeneous with *P. calcabudae* (Table S1). According to earlier studies, however, *P. restrictum* was not limited to the coastal areas. Furthermore, there are few reports on the environmental distribution of *P. calcabudae*. *Aspergillus terreus* was identified in six host species (Table S1). This symbiotic endophyte triggers host resistance to phytopathogens and has direct antagonistic activity [Halo]. It is also a fungal symbiont in insects that spread pathogens and pose physical threats to a wide range of host plants [17]. The distribution of *Talaromyces australis* has only been reported for Australia wherein it was first discovered [62]. It resembled the culture collection isolated from the eight species of marine plants in the present study. Sixteen isolates from the wooded perennials *P. thunbergii* and *R. rugosa* showed molecular similarity to this species. However, it cannot be said that only these dominant fungi are important for sand dune restoration.

Endophytes that emerge in response to the ecological pressure of the harsh ambient environment will be highly selective. Furthermore, the range of endophyte species that emerge may also be affected by the host dune plant species themselves. However, the probability that the host plants will benefit from symbioses may increase with endophyte diversity interactively. Endophytes such as *Mortierella* have been reported in extreme marine terrains [63,64], but were not detected here.

Various species belonging to the genus *Fusarium* are widely known to cause plant diseases or cause fungal toxins in food plants. It is also reported as an endophytic fungus in various plant species [65]. However, in contrast, various *Fusarium* species belonging to endophytic fungi are sometimes reported as non-pathogenic. Such non-pathogenic *Fusarium* isolates rather provide resistance to plant infectious diseases [66,67]. It also induces the growth of plants through specific ecological relationships [68]. As a result,

the genus Fusarium cannot be seen as negative on the basis of being identified in this study, and further study may be possible on how *Fusarium* as an endophyte has developed interactions with preserved coastal plants for a long time.

### 3.3.3. Uniqueness of Non-Dominant Fungal Genera

The 13 fungal genera identified here were not heretofore reported for the marine environment or root colonization by halophytes. These included the non-dominant fungal genera *Ramichloridium* (isolated from *Z. sinica*), *Rhexocercosporidium* (isolated from *L. japonica*), *Neophaeothecoidea* (isolated from *Setaria viridis*), *Antennariella* (isolated from *Artemisia scoparia*), *Pseudopyrenochaeta* (isolated from *S. kamtschaticum, P. camtschatica*), *Phialomyces* (isolated from *P. thunbergii*), *Sagenomella* (isolated from *I. anthephoroides*), *Mycofalcella* (isolated from *P. camtschatica*), *Collembolispora* (isolated from *A. sphathulifolius*), *Arcopilus* (isolated from *P. thunbergii*), *Melanconium* (isolated from *S. marina*), *Pezicula* (isolated from *A. sphathulifolius*), and *Acremoniopsis* (isolated from *P. thunbergii*). The halophyte species *Z. sinica, L. japonica, S. kamtschaticum, I. anthephoroides*, and *S. marina* were symbiotic with the foregoing root-colonizing fungal genera, and their endophyte clusters were heretofore unreported. The halophytes *S. marina* and *Z. sinica* have been utilized in genetic and physiological research [69–71] as well as landscape retention [19]. Investigation of the interactions between the endophytes and their host coastal plant species might disclose novel biological resources.

The probability of successful restoration of disappearing coastal dunes might increase with plant diversity. The present study identified all endophytic fungi in all coastal plant species contributing to phyto-diversity and ecosystem stability in coastal dune vegetation. We detected various fungal genera previously unreported as marine-derived or coastal plant endophytes. These novel taxa accounted for 30% of the 39 fungal genera identified.

They might play important roles in enhancing endophyte cluster diversity and stability. The disappearance of coastal dunes and salt wetlands and the simplification of coastlines are emerging as serious problems. The main causes are artificial destruction, natural erosion, and rising sea levels due to climate change and global warming [10]. In order to solve this problem, native coastal plants or halophyte species that have evolved and adapted in a well-preserved coastal sand dune environment can be the answer. The stable prosperity of native plants vegetation prevents sand loss in coastal dunes, preventing the loss of coastal topology [3]. To do so, more diverse endophyte culture collections are required that demonstrate long-term interaction with their coastal halophytes while enduring extreme marine environments. In contrast, in terms of securing biological resources, securing as many diverse microbial resources as possible can be one goal [17]. For this reason, genetic diversity investigations and culture collection of symbiotic microorganisms are being conducted in a special environment or in higher organisms that are native to a unique natural terrain [17]. It is related to the entry into force of the Nagoya Protocol to secure the maximum variety of biological resources [17]. Since the introduction of the Nagoya Protocol, securing biological resources has been accelerating, and one of the purposes is also related to solutions for diverse environmental problems using biological resources [72].

### 3.4. Fungal Endophyte Diversity

#### 3.4.1. Pinus Thunbergii Endophytes

Endophyte diversity was evaluated according to genera richness (S) (10–2), Margalef's richness (Dmg) (3.057–0.621), Menhinick's index (Dmn) (2.449–0.894), Shannon's diversity (H') (1.972–0.500), and Simpson's index (D) (0–0.600).

The dune plant community comprised two annuals, three biennials, six perennial herbs, two perennial shrubs, and a perennial tree. Stratification was not observed in the Hwajin sand dune community. The *P. thunbergii* canopy increased with distance from sea level.

There was little variation in herbaceous plant diversity. By contrast, *P. thunbergii* displayed the highest morphological diversity (Figure 3) and formed a thick canopy.

*Pinus thunbergii* trees thrive for extended periods at the rear of coastal sand dunes and retard dune erosion. The distribution patterns of their endophytes markedly differ from those of herbaceous plants. *Talaromyces* predominates in *P. thunbergii* (Figure 3). By contrast, *Penicillium* was extremely dominant in the herbaceous plants (Figure 3). As *Talaromyces* predominated, the Simpson's index was not low (Figure 3).

### 3.4.2. Annual, Biennial, and Perennial Plant Endophytes

Comparisons among annual, biennial, and perennial plants disclosed no specific patterns of variation in diversity. This finding was significant as it was believed that the probability of fungus endogenization would increase with the duration of the plant settlement period. In fact, this hypothesis was demonstrated to be invalid. Most coastal halophytes thrive in colonies confined to specific sites and their populations are stable over long periods of time. The single individual that composes the annual halophyte colony lives for only one to two years, but the colony's settlement period can be seen as lasting for much longer because many halophyte individuals gather. So long as their habitat conditions are well preserved, even if their survival period is short, they can nonetheless interact with a wide range of fungal species as they continuously colonize the same sites. This observation suggests that the Hwajin coastal sand dunes and the plant communities that they comprise are well preserved and form a stable climax community. Since individual plants and colonies of individual plants might establish differentiated symbiotic relationships with microorganisms, comparative analytical studies on two such cases may yield interesting results.

### 3.4.3. Penicillium Dominance and Diversity

When the *Penicillium* dominance ratio decreased in a specific host (Figure 3), the diversity and Margalef's richness of other fungal genera increased (2.171 for *C. dactylon*, 2.919 for *Carex pumila*, 2.791 for *P. australis*, 3.057 for *P. camtschatica*, and 2.404 for *Z. sinica*) (Figure 3; Table 3). As a result, *Penicillium* might encounter ecological competition from members of other fungal genera. Concurrently, *Penicillium* may adapt to the marine environment or the host plants interacting with it. When *Penicillium* shows high dominance, the morphological diversity of the endophytes declines in the host species. Plots of total isolates (morphological diversity), changes in Margalef's richness, genera richness, and morphological diversity and abundance showed similar trends.

By contrast, morphological diversity was very high while genera richness (S') and Margalef's richness were low in *Atriplex gmelinii* (Figure 3). *Penicillium* predominated in *A. gmelinii* (Figure 3). The wide diversity of *Penicillium* isolates reduced the richness of the genus and increased Simpson's indices.

As of now, 40% of the coastline has been reduced on the west coast of the Korean Peninsula due to the destruction of coastal dunes [9,28,29], and 50% of the coastline has been shortened worldwide [10]. The secured endophyte strains and diversity information about native coastal plant species and their endophytes can be used in two main directions: first, it is possible to test at the laboratorial level whether secured strains affect the survival of host plants in terms of plant growth promotion or ISR activity. Cultures whose effects have been verified may be applied to the coastal site to promote the prosperity of native plants in coastal dunes. It is necessary to induce the prosperity of native or planted coastal vegetation to prevent erosion of coastal dunes, marsh wetlands, or coastlines, and a study about the Ramsar wetlands located on the Korean Peninsula demonstrated that the stable prosperity of coastal plants enables the expansion of coastal wetlands [10]. Second, information on the microbial diversity of endophytic fungi identified in advance in healthy coastal dunes can be an indicator of whether coastal plant clusters have a healthy symbiotic relationship with their endophyte clusters when restoring coastal dunes in the future.

## 4. Conclusions

The aim of this study was to determine the distribution and diversity of fungal endophytes in coastal plants native to natural sand dunes that have been effectively protected from anthropogenic activity. A total of 361 fungal endophyte species in 39 genera were identified and compiled into a culture collection that could be used to restore damaged coastal dunes in the future.

In addition to expanding access to biological resources and enhancing the knowledge of their diversity and distributions, there are two important implications of this study. First, future coastal sand dune conservation studies should examine the biological resources of entire bioclusters and not merely the dominant plants or their endosymbionts. Second, in well-preserved coastal dunes, the physiological lifespan of herbaceous coastal plants and the diversity of endogenous fungi are not proportional. This suggests the stable settlement of coastal plant populations.

**Supplementary Materials:** The following are available online at https://www.mdpi.com/article/10.3390/jmse11040734/s1, Figure S1: Phylogenic relation of endophyte fungi in each host coastal plant species, Table S1: Fungal isolates and their taxonomic identity from each coastal plant species.

**Author Contributions:** Conceptualization, Y.-H.Y.; project administration, Y.-H.Y.; data curation, J.M.P.; formal analysis, J.M.P.; funding acquisition, Y.-H.Y.; project administration, Y.-H.Y.; software, J.M.P.; validation, Y.-H.Y.; writing—original draft, J.M.P.; visualization, J.M.P.; writing—review and editing, J.M.P. All authors have read and agreed to the published version of the manuscript.

**Funding:** This work was supported by a grant from the National Institute of Biological Resources (NIBR), funded by the Ministry of Environment (MOE) of Korea (Nos. NIBR202315101).

**Institutional Review Board Statement:** Not applicable.

**Informed Consent Statement:** Not applicable.

**Data Availability Statement:** All the raw sequences obtained from this study were deposited at the GenBank of the National Center for Biotechnology Information (NCBI) with assigned accession numbers JX238686–JX239046 (361 sequences).

**Conflicts of Interest:** The authors declare no conflict of interest.

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
