# Peer review of "Culturable Endophyte Fungi of the Well-Conserved Coastal Dune Vegetation Located on the East Coast of the Korean Peninsula"

_jmse, doi:10.3390/jmse11040734_

Round 1

Reviewer 1 Report

The study presented in this manuscript consists of an observational study of fungal endophytes in vascular plants of the back-dune region of a beach along the east coast of the Republic of Korea. While there is a long history of studying mycobiota associated with plants, much of this work was conducted using agar and other culture media to sample fungal diversity, as well as much of this work was focussed on a few plant species, often in agricultural settings. With the growth of DNA sequence analysis as an efficient tool for examining fungal communities, and increasing awareness of the importance of the micro- and mycobiota on animals and plants, this study has the potential to be of broad interest. I do have a couple of questions that I would need to see addressed, but in general the background and goals of the study are presented clearly, and the methods generally are reasonable and appropriate.

My specific questions and comments about the methods and results are:

First, the primer pair listed is ITS1 and ITS4, which are commonly used in fungal community assessment, as these primer pairs span ITS1, 5.8S and ITS2 regions. However, the Methods section indicates that only a partial ITS1 region sequence was obtained, while primer ITS1 binds to the small subunit (18S), so complete ITS1 region sequences should have been obtained. I may have missed the explanation for this in the Methods; in either case, if the authors could clarify this issue, that would help evaluate the Results.

Somewhat along those lines, ITS1F usually is the preferred primer for fungal barcoding, as it is more specific to fungi (though, also more specific to asco- and basidiomycetes). If it could be clarified why ITS1 rather than ITS1F was used as the forward sequence, that also would help evaluate the Results. (Thank you for including the primer sequences, as this helped confirm the primer pair chosen.)

For the Results, the data presented clearly support the assertion that Pinus thunbergii had the most diversity fungal community, at least at the level of genera. However, the Abstract states that the fungal community on P. thungergii consisted exclusively of the genus Talaromyces. This may be simply a grammatical issue, but if it could be clarified, that should resolve the conflicting information.

Also, while the Results/Discussion briefly address the issue of pathogens being included within the fungal species observed (paragraph beginning l. 281), this portion of the text doesn’t address that at least some of the fungal species (e.g., Fusarium), most likely are pathogens.

On the other hand, although mycorrhizal fungi would be expected to be included in the fungal endophytes detected in these samples, this issue isn’t directly addressed.

More generally, the Abstract and main text of the manuscript don’t emphasise that only roots were sampled, and the overall endophytic community would be more complex if the entire plant was sampled.

Finally, while the study focussed on genus-level diversity, the sequences seemed to generally match well to individual species (supplemental docs), and these primer pairs generally are pretty robust at the species level, which is why they’re commonly used. (As the authors note, though, there are some inconsistencies and problems with sampling the entire diversity of fungi.)

Author Response

Dear reviewer, please see an attached file ("author response")

Reviewer 2 Report

Dear Authors,

Here are my comments on your manuscript:

- Fig. 2S - must be changed, as the names are tough to read

- please remove the space between ° and C (lines: 110, 118, 125-127)

- the names of the various levels of the classification (like class, phylum, and order) should not be capitalised. Please change it (lines: 162, 168, 169,227,252).

Good luck with your work!

Reviewer

Author Response

(The authors gave the same response as above.)

Reviewer 3 Report

The rationale for the study is not clear, hence my ratings above.

Much is known about fungi in sand dunes and there is no mention of any of these. Why are the fungi associated with sand not found as endophytes.

Why are all the fungi listed just generalist taxa?

These are worrying features of studies on endophytes, and why re certain groups of fungi rarely found as endophytes e.g basidiomycetes?

Hence I rate this low

Author Response

(The authors gave the same response as above.)

Reviewer 4 Report

The article analyzed the diversity of culturable endophytic fungi of coastal dune vegetation on the Korean Peninsula. A total of 361 endophytic fungi belonging to 7 classes and 39 genera were isolated from 24 halophyte species, of which 13 genera had not been previously reported as marine environmental fungi. The results broadened the understanding of the endophytic fungi of coastal plants, disclosed several heretofore unreported associations between endophytes and coastal plants, but there were some problems with the manuscript:

1.In this paper only endophytic fungi were isolated and only one culture medium was used. Since the article did not perform high-throughput sequencing, it is suggested that the title be limited to ‘culturable endophytic fungi’.

2. Lin 19-20, in the abstract, mentioned that ‘Pinus thunbergii consisted exclusively of Talaromyces’, but it appeared from the results that both Table 3 and Figure 3 showed that Pinus thunbergii contains more than one genus. How can this result be interpreted?

3.The literature cited in the introduction is too old, with only five citations in the last five years. The current challenges to coastal environmental protection and development are mentioned extensively, but specific solutions still need to be addressed, and the importance of coastal plant endophytes needs to be highlighted.

4. The colours in the legend for Margalef’s richness in Figure 3 do not correspond to those actually used in the figure. Please check.

5. The role of endophytic fungi was also not verified by the re-inoculation experiment either, and all analyses are based on speculation, how should sand dune restoration be carried out? How can we use the symbiotic relationship between endophytic microorganisms and plants? The authors should explain this.

6.The results and discussion also do not focus on the significance of the endophytic fungi’s possible presence, which have not been previously reported in the marine environment. Please add to the discussion.

Author Response

(The authors gave the same response as above.)
